# Phonon-driven spin-Floquet magneto-valleytronics in MoS$_2$

Dongbin Shin [1], Hannes Hübener [2], Umberto De Giovannini [2], Hosub Jin[1], Angel Rubio [2,3,4] & Noejung Park [1,2]

Two-dimensional materials equipped with strong spin–orbit coupling can display novel electronic, spintronic, and topological properties originating from the breaking of time or inversion symmetry. A lot of interest has focused on the valley degrees of freedom that can be used to encode binary information. By performing ab initio time-dependent density functional simulation on MoS$_2$, here we show that the spin is not only locked to the valley momenta but strongly coupled to the optical $E''$ phonon that lifts the lattice mirror symmetry. Once the phonon is pumped so as to break time-reversal symmetry, the resulting Floquet spectra of the phonon-dressed spins carry a net out-of-plane magnetization ($\approx 0.024\mu_B$ for single-phonon quantum) even though the original system is non-magnetic. This dichroic magnetic response of the valley states is general for all 2H semiconducting transition-metal dichalcogenides and can be probed and controlled by infrared coherent laser excitation.

[1] Department of Physics, Ulsan National Institute of Science and Technology, UNIST-gil 50, Ulsan 44919, Korea. [2] Max Planck Institute for the Structure and Dynamics of Matter Center for Free-Electron Laser Science, Luruper Chaussee 149, Hamburg 22761, Germany. [3] Center for Computational Quantum Physics (CCQ), The Flatiron Institute, 162 Fifth Avenue New York, New York, NY 10010, USA. [4] Nano-Bio Spectroscopy Group, Departamento de Fisica de Materiales, Universidad del País Vasco UPV/EHU, San Sebastián 20018, Spain. Correspondence and requests for materials should be addressed to A.R. (email: angel.rubio@mpsd.mpg.de) or to N.P. (email: noejung@unist.ac.kr)

Spin manipulation of charge carriers[1–3] and the controllable switching of a few-atom magnetic unit[4,5] have attracted a lot of attention in recent decades[6,7]. Spin–orbit coupling (SOC) is the core ingredient enabling the control of these effects[8], as it provides a tunable intrinsic magnetic field through the change of the scalar electronic potential. In parallel, optically induced ultrafast spin dynamics have been studied in detail[9–11]. Low-frequency terahertz sources are expected to have particular relevance here as they allow for a straightforward control avoiding high energy transfer to the material[9]. The spin dynamics can be coupled to phonons via the SOC interaction and a coherent infrared (IR) laser excitation can be used to control phonons and thus to modify the effective gauge-field felt by the electronic spin[12]. In two-dimensional (2D) semiconductors with strong SOC time-reversal symmetry partners often constitute valleys with very distinct electronic and spin properties. In fact, these valleys are characterized by a strong electronic spin–momentum locking that can be used to encode binary information, known as valleytronics[13–15]. To achieve a controllable asymmetry in the valleys of transition-metal dichalcogenides (TMDCs), some recent studies have used either a static magnetic field[16–19] or the optical Stark effect[20,21].

Here, we focus instead on phonon-dressed spin-valley states. Using MoS$_2$ as test material, we explore the dynamic evolution of spins at the K- and K′-valleys when a coherent phonon mode is excited with a weak laser pulse. We show that, while the spins of the valence band maxima (VBM) are largely frozen, those on the conduction band minima (CBM) exhibit interesting dynamics governed by one particular optical phonon. We perform extensive ab initio real-time electron-ion propagation within time-dependent density functional theory (TDDFT)[22,23]. As a result, we find that the full spin dynamics in the valley of MoS$_2$ is well described by a simple two-level Hamiltonian in which the internal magnetic field oscillates along the particular optical phonon that breaks the in-plane mirror symmetry[12]. We show that the Floquet state for the valley-locked spin is characterized by two distinct

Larmor precessions whose amplitude is determined by $\sqrt{n_{ph}+1}$, where $n_{ph}$ is the number of the phonon excited in the system by the external laser field.

## Results

**Phonon-induced spin dynamics calculated within TDDFT.** In the TDDFT calculation, the two-component Kohn–Sham spinors $|\psi_{n,\mathbf{k}}\rangle$ evolve following the time-dependent Kohn–Sham equation:

$$
i\hbar \frac{\partial}{\partial t}|\psi_{n,\mathbf{k}}\rangle = \left[\left(-\frac{\hbar^2}{2m}\nabla^2 + \sum_\lambda \hat{v}_{pp}(\mathbf{r}-\mathbf{R}_\lambda(t))\right) + v_{Hxc}[\rho(t)] + \mu_B \hat{\boldsymbol{\sigma}} \cdot \frac{\partial E_{xc}}{\partial \mathbf{m}} + \hat{v}_{SOC}\right]|\psi_{n,\mathbf{k}}\rangle,
\tag{1}
$$

where $\hat{v}_{SOC} = \frac{\hbar}{4m^2c^2}\hat{\boldsymbol{\sigma}} \cdot (\boldsymbol{\nabla}V \times \hat{\mathbf{p}})$ is the spin–orbit potential with $V(\mathbf{r})$ representing the sum of the local part of the atomic potential $\hat{v}_{pp}(\mathbf{r})$ and the Hatree-exchange-correlation potential $v_{Hxc}(\mathbf{r})$. The magnetization vector field is defined as $\mathbf{m}(\mathbf{r}) = \mu_B \sum_{n,\mathbf{k}} \psi_{n,\mathbf{k}}^+(\mathbf{r})\hat{\boldsymbol{\sigma}}\psi_{n,\mathbf{k}}(\mathbf{r})$ with $\mu_B$ indicating the Bohr magneton ($\mu_B = e\hbar/2m$). The ion dynamics $\mathbf{R}_\lambda(t)$ follows the classical Newton equation $M_\lambda \frac{d^2}{dt^2}\mathbf{R}_\lambda(t) = \mathbf{F}_\lambda(t)$ with the instantaneous Ehrenfest forces acting on each ion (Methods). The time-dependent spin and charge densities are computed directly from the time-dependent Kohn–Sham spinors as: $\mathbf{S}_{n,\mathbf{k}}(t) = \langle \psi_{n,\mathbf{k}}(t)|\frac{\hbar}{2}\hat{\boldsymbol{\sigma}}|\psi_{n,\mathbf{k}}(t)\rangle$ and $\rho(t) = \sum_{n,\mathbf{k}} \psi_{n,\mathbf{k}}^*(t)\psi_{n,\mathbf{k}}(t)$. Detailed description of the computational parameters used for the DFT[22,24] and TDDFT[23] are given in Supplementary Note 1.

To start the TDDFT simulations, we mimic the effect of a resonant right-handed photon by promoting one spin down electron from the VBM to the CBM of the K valley, as schematically depicted in Fig. 1a. Then, we monitor the electron dynamics of this excited state when different zone-center phonon modes are coherently excited. To simulate the induced dynamical

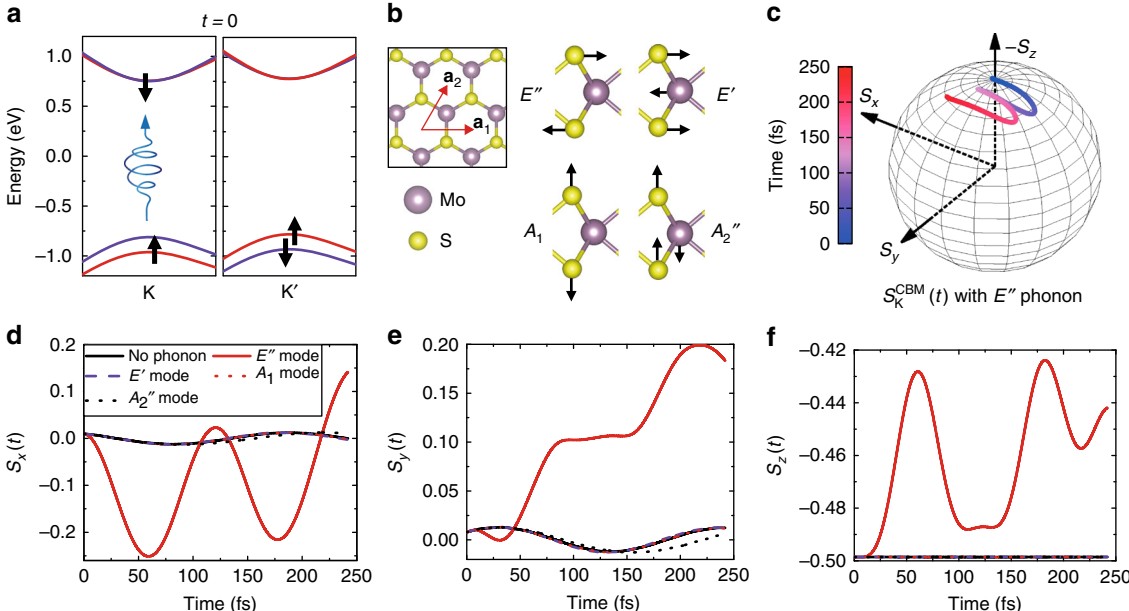

**Fig. 1** TDDFT simulation of the spin-phonon dynamics of monolayer MoS$_2$. **a** The initial electronic configuration used here with one electron excited at the K valley, for example, by right-handed light. **b** Top view of the solid and a side view of the eigenvectors of the four zone-center optical phonons. **c** The time trace of the spin vector of the CBM electron with the $E''$ phonon being coherently excited. **d–f** The same time profiles of the Cartesian components of the spin with each of the phonon modes $E''$, $E'$, $A_1$, $A_2''$ and with the frozen lattice. The full phonon dispersion is given in Supplementary Fig. 1. Here, all phonons are linearly polarized along the $y$ direction

effects of the phonons, we initialize each atom in its equilibrium position with a finite velocity along the normal mode of each phonon we want to excite. The shape of each phonon mode is depicted in Fig. 1b. The calculated time profiles of the spin of the excited electron at the CBM of the K valley are presented in Fig. 1c–f. We highlight that only the $E''$ optical phonon mode appreciably couples to the spin motion, while the other three phonons ($E'$, $A_1$, and $A_2''$) are basically uncoupled. This particular feature can be related to the fact that the $E''$ is the only optical vibration that breaks both the mirror and trigonal symmetries of the $MoS_2$ plane, while others preserve at least one of the symmetries. The spin profile affected by the $E''$ mode is shown on the Bloch sphere in Fig. 1c. This spin motion is neither in plane nor out of plane and has a precession that turns out to be proportional to the amount of phonon displacement. The time traces of the Cartesian components of the spin driven by each optical phonon mode are presented in Fig. 1d–f. As a reference, we show the calculated spin dynamics for the frozen equilibrium ionic configuration (black curve) starting from the same excited electronic initial condition (Fig. 1a). The spin dynamics with the latter three phonons ($E'$, $A_1$, and $A_2''$) are almost the same as that in the frozen lattice configuration. Since the excited electronic configuration at $t=0$ (Fig. 1a) deviates from the electronic self-consistent ground state of the material, the electron exhibits a minor dynamics even with the frozen lattice. Moreover, the spins in the VBMs mostly remain near the equilibrium configuration during the time evolution, which can be inferred from the fact that their intrinsic up/down spin splitting is an order of magnitude larger than the phonon-induced in-plane SOC magnetic field (given in Supplementary Figs. 2d and 3d).

**Static DFT calculations of the phonon-induced magnetism.** We now formulate a minimal model Hamiltonian that captures the

essence of the aforementioned spin dynamics. SOC splits the bands, as presented in Fig. 2a, except at the symmetry-protected degenerate points[14,25]. The up/down splitting of the CBM and the VBM near K and K′ amount to 3 and 156 meV, respectively[14,15,26]. The monolayer $MoS_2$ honeycomb structure has a mirror symmetry plane at the central Mo layer which in the point group of $D_{3h}$ enforces the spins to be aligned out of the plane. All these features can be cast into a simple two-level Hamiltonian. For the CBM state in the K and K′ valleys, the up/down energy separation can be modeled as a Zeeman splitting induced by an effective magnetic field of the form: $\mathbf{B} = \tau B_0 \hat{\mathbf{z}}$, where $\tau = 1$ and $\tau = -1$ for K and K′, respectively. The corresponding model Hamiltonian reads as $\hat{H} = \frac{e}{m} \hat{\mathbf{S}} \cdot \mathbf{B} = \tau \varepsilon_0 \hat{\sigma}_z$, with an energy parameter $\varepsilon_0 = 1.5$ meV (fitted to our first-principles calculations). The observed dynamical effect of the phonon is accounted for by this effective Hamiltonian via including an additional effective magnetic field that mimics the spin–phonon coupling, namely $\mathbf{B}(t) = \tau B_0 \hat{\mathbf{z}} + \mathbf{B}_{ph}(t)$. The SOC potential ($\hat{v}_{SOC}$) in Eq. (1) reveals how the effective magnetic field $\mathbf{B}_{ph}(t) = \frac{1}{2m^2c^2} \frac{\partial}{\partial \mathbf{r}} V[\mathbf{R}_\lambda] \times \hat{\mathbf{p}}$ appears as a result of the atomic motion $\mathbf{R}_\lambda(t)$.

To quantify the phonon dependence of the effective Hamiltonian, instead of dealing with the operator form for $\mathbf{B}_{ph}(t)$, here we directly compute the spin resolved electronic structure variations induced by the static atomic displacements following each phonon mode near the CBM at the K valley. The results for the $E''$ phonon and the other three optical phonons are shown in Fig. 2b, d, respectively. Except for the $E''$ phonon, all the other phonon modes do not change the spin texture of the bands around K. Similar results are obtained near K′ but not shown. In fact, for static displacement along the selected $E''$ eigenmode the spin of the CBM lies almost in the $x$–$y$ plane along the $y$ direction. In Fig. 2c, we show the inclination angle of the spin vector and the up/down splitting gap ($\Delta \varepsilon$) of the CBM as a function of the

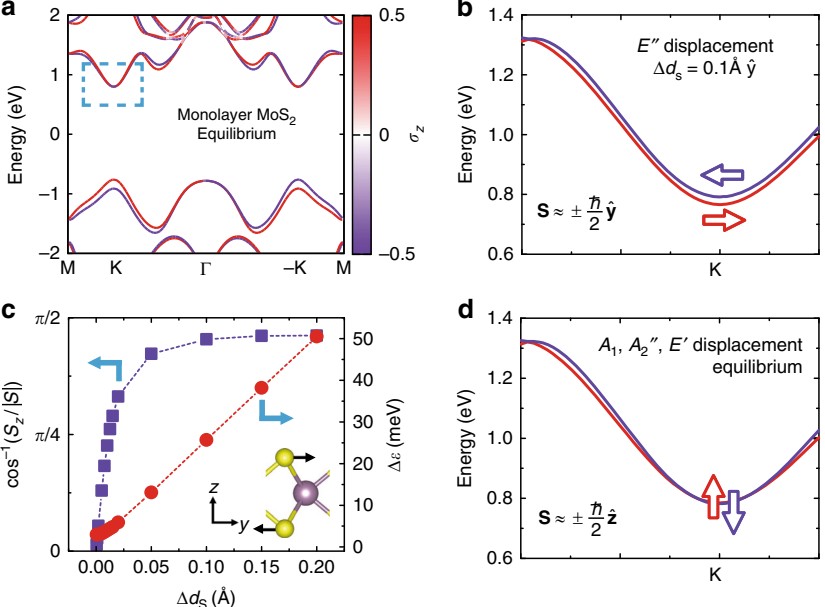

**Fig. 2** Electronic structures of monolayer $MoS_2$ in equilibrium geometry and with the displacement along a specific coherent phonon mode. **a** The band structure with the equilibrium geometry. **b** Zoomed-in view of the CBM near the K valley with a displacement along $E''$. **c** The variations in the inclination of the spin angle and the up/down splitting ($\Delta \varepsilon_{gap}$) with respect to the magnitude of the displacement along the $E''$ phonon mode. **d** Zoomed-in view of the CBM near the K valley with the equilibrium geometry and with the displacement along the other three zone-center phonon modes $E'$, $A_1$, $A_2''$. In **b**, **d** the displacement vectors are normalized such that the maximum shift of the atomic position is 0.1 Å. The dashed box in **a** indicates the window for the zoom in **b** and **d**. Inset in **c** depicts the direction of the atomic displacement in $y$ direction. Further details of the role of the $E''$ phonon mode are given in Supplementary Table 1 and Supplementary Figs. 3 and 4

magnitude of the atomic displacement along the phonon mode. The angle for the spin vector is defined by $\theta = \cos^{-1}(S_z/S)$, where $S$ is the norm of the vector, i.e., $\hbar/2$. For this plot, we note that the spin is gradually canted towards the $y$ direction as the atomic displacement increases and the $z$-component ($S_z$) is reduced but remains finite over the range of displacement shown in Fig. 2c. These variations in the spin structures can be modeled by a magnetic field in $y$ direction, as $\mathbf{B} = \tau B_0 \hat{\mathbf{z}} + B_{\mathrm{ph}} \hat{\mathbf{y}}$. We note that the $E''$ mode is doubly degenerate at the zone center of the phonon Brillouin zone, and thus the linear combination of the two eigenvectors can be chosen such that the in-plane component of the induced spin points in any direction (summarized in Supplementary Table 1). Furthermore, we want to emphasize that exciting a superposition of two linear $E''$ modes in different directions can result, depending on the relative detuning of the two modes, in a circular or elliptically polarized phonon with the same frequency.

Since the lattice distortion along the $E''$ phonon mode creates a net effective in-plane magnetic field while the other three phonons are almost ineffective, the previously derived two-level Hamiltonian will inherit the time profile of the $E''$ phonon mode, i.e., $\hat{H}(t) = \hat{H}(t + 2\pi/\omega_{\mathrm{ph}})$, where $\omega_{\mathrm{ph}}$ the frequency of the $E''$ mode. This Hamiltonian, being perfectly periodic in time and describing the dynamics of our spin–phonon-driven system, suggests that the spin states can be described in terms of the corresponding Floquet spectrum, as will be outlined below[27]. The details of the model Hamiltonian studies are given in Supplementary Notes 2 and 3.

**Simplified model hamiltonian.** To illustrate this new phonon-mediated spin-Floquet non-equilibrium state of the material, we explicitly incorporate the time oscillation of the phonon into the model Hamiltonian in the form of a simple trigonometric function. As for Fig. 1, the phonon is assumed to oscillate along the $y$ direction, producing a magnetic field along the same direction of $\mathbf{B}_{\mathrm{ph}}(t) = B_{\mathrm{ph}} \sin(\omega_{\mathrm{ph}}t)\hat{\mathbf{y}}$. The corresponding two-level

Hamiltonian becomes $\hat{H}(t) = \varepsilon_0 \hat{\sigma}_z + \varepsilon_{\mathrm{ph}} \hat{\sigma}_y \sin(\omega_{\mathrm{ph}}t)$, where $\varepsilon_{\mathrm{ph}} = \frac{e\hbar}{2m} B_{\mathrm{ph}}$ and the effective magnetic field $B_{\mathrm{ph}}$ depends implicitly on the amplitude of the phonon mode. The time evolution of the two-component state vector is calculated by $|\psi(t + \Delta t)\rangle = \exp\left(-\frac{i}{\hbar}\hat{H}(t)\Delta t\right)|\psi(t)\rangle$. The spin dynamics is presented in Fig. 3a. We note that the calculated spin trajectories are in good agreement with the full ab initio TDDFT simulation performed shown in Fig. 1 for a shorter time interval up to $2T$=244 fs. This excellent performance of the simple $2 \times 2$ model allows for an accurate description of the spin dynamics for very long times, which would be difficult to reach with the first principles TDDFT approach. The plot in Fig. 3a shows only the case of $\varepsilon_{\mathrm{ph}}$=$3\varepsilon_0$, however we verified that we get qualitatively the same behavior for a set of different values of $\varepsilon_{\mathrm{ph}}$.

**Floquet analysis and phononic dichroism.** The field of spintronics is evolving at very high speed[28]. In particular, the possibility of realizing light control of the spin has attracted huge interest as a way to seamlessly bridge magnetic responses and spin manipulation[4,5,10,11]. To achieve this goal, it was proposed (e.g., in refs. [11,29]) to use a time-dependent circularly polarized perturbation (phonon or photon) to switch the angular momentum eigenstate of the material[11,29]. Although they indicate that the spin configuration of magnetic materials can be indeed controlled by light, the non-equilibrium magnetic response in an optically driven state of a non-magnetic material has not yet been demonstrated. The open question in this regard is whether a non-magnetic material, for instance a semiconducting 2D material, can be driven into a stationary magnetic state by a time-reversal breaking perturbation. We address this point in detail next. To illustrate this concept, we take the aforementioned two-level Hamiltonian and extend it to the case where the driving is a circularly polarized phonon. In this case the time-dependent Hamiltonian reads $\hat{H}(t) = \varepsilon_0 \hat{\sigma}_z + \varepsilon_{\mathrm{ph}}\left(\hat{\sigma}_x \cos(\omega_{\mathrm{ph}}t) - \hat{\sigma}_y \sin(\omega_{\mathrm{ph}}t)\right)$. The time-dependent Schrödinger equation can be solved analytically using

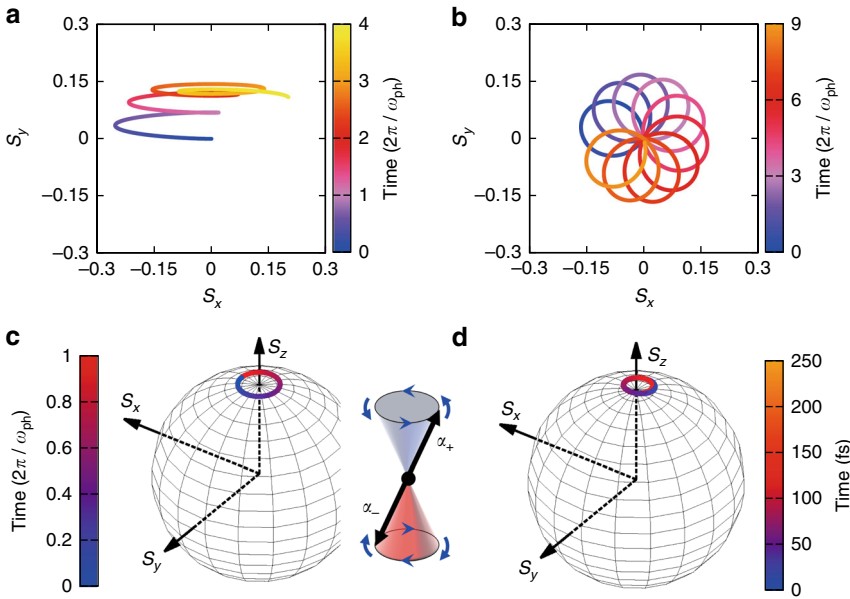

**Fig. 3** Time evolution of the spin driven by the $E''$ mode. **a**, **b** Model Hamiltonian calculation of the spin trajectory starting at $t$=0 with the spin-down state shown in Fig. 1a with **a** a linearly polarized and **b** a circularly polarized in-plane magnetic field. **c**, **d** The Floquet eigenstate on the Bloch sphere for **c** the model Hamiltonian with the circularly polarized in-plane magnetic field and **d** the full TDDFT solution with the circularly polarized $E''$ phonon. The inset of **c** depicts schematically the two Floquet eigenstates, $\alpha_+$ and $\alpha_-$ described in the main text. In **a**–**c** the time scale is presented in units of the $E''$ phonon period, i.e., $T$=$2\pi/\omega_{\mathrm{ph}}$=122 fs

the rotating-wave approximated model for Rabi oscillation[30,31]. The spinorial solution is

$$|\Psi(t)\rangle = \begin{pmatrix} -ie^{i\frac{\omega_{ph}}{2}t}\frac{\varepsilon_{ph}}{\hbar\Omega}\sin(\Omega t) \\ e^{-i\frac{\omega_{ph}}{2}t}\frac{i\Delta\sin(\Omega t)+\Omega\cos(\Omega t)}{\Omega} \end{pmatrix}, \text{where}$$
$$\Delta = \frac{\omega_{ph}}{2}+\frac{\varepsilon_0}{\hbar} \text{ and}$$
$$\Omega = \sqrt{\Delta^2+(\varepsilon_{ph}/\hbar)^2}. \tag{2}$$

The corresponding spin trajectory is presented in Fig. 3a, b for both linearly and circularly polarized phonons, respectively. The spin trajectories exhibit complicated femto-seconds dynamical features, and the spin vector is not restored to its original position unlike the Larmor-type spin precession induced by a static magnetic field. As is the case of a two-level fermionic system coupled to a bosonic oscillator[31], the level spacing ($\varepsilon_0$), the frequency ($\omega_{ph}$) and the amplitude ($\varepsilon_{ph}$) of the perturbation are all intertwined. However, here the vector-nature of the spin produces more structure than the simple two-level Fermionic oscillation. These complex spin dynamics can be rationalized in terms of Floquet states[27,32]. For a given phonon ($\omega_{ph}$) driven Hamiltonian, the Floquet states are quasi-stationary spinors $|\Psi_\alpha\rangle = e^{-i\alpha t}|\Phi_\alpha\rangle$ that satisfy the following equation.

$$\left[\hat{H}(t)-i\hbar\frac{\partial}{\partial t}\right]|\Phi_\alpha\rangle = \alpha|\Phi_\alpha\rangle, \text{ with } |\Phi_\alpha(t)\rangle = |\Phi_\alpha(t+2\pi/\omega_{ph})\rangle. \tag{3}$$

This $2\times2$ matrix eigenvalue equation is also exactly solvable as discussed above[30]. The only difference for this case is that we have to impose periodic boundary condition in time, instead of the fixed initial condition used to get the solution shown in Eq. (2). The corresponding eigenvalues and eigenvectors can be written as

$$|\Psi_{\alpha_+}\rangle = e^{-i\alpha_+ t}|\Phi_{\alpha_+}\rangle$$
$$= \frac{\varepsilon_{ph}/\hbar}{\sqrt{2\Omega(\Omega-\Delta)}}e^{-i\left(\frac{\omega_{ph}}{2}+\Omega\right)t}\begin{pmatrix} e^{i\omega_{ph}t} \\ \frac{\Omega-\Delta}{\varepsilon_{ph}/\hbar} \end{pmatrix}, \text{with}$$
$$\alpha_+ = \frac{\omega_{ph}}{2}+\Omega.$$
$$|\Psi_{\alpha_-}\rangle = e^{-i\alpha_- t}|\Phi_{\alpha_-}\rangle$$
$$= \frac{\varepsilon_{ph}/\hbar}{\sqrt{2\Omega(\Omega+\Delta)}}e^{-i\left(\frac{\omega_{ph}}{2}-\Omega\right)t}\begin{pmatrix} -e^{i\omega_{ph}t} \\ \frac{\Delta+\Omega}{\varepsilon_{ph}/\hbar} \end{pmatrix}, \text{with}$$
$$\alpha_- = \frac{\omega_{ph}}{2}-\Omega. \tag{4}$$

These two states are always orthogonal to each other, and the spin expectation of one is opposite to the other, as depicted in the inset of Fig. 3c. Each Floquet state is characterized by its rotation with respect to the fixed axis perpendicular to the 2D plane, possessing a time-independent constant perpendicular component of the spin vector, namely a fixed $S_z$ value, like in a typical Larmor precession. These two Floquet eigenstates span the whole SU(2) space for spinors at any time, and any dynamical spin motion can be resolved and analyzed in terms of this basis.

To corroborate the conclusions from the model spin dynamics discussed above, we now turn back to the first-principles materials simulation to address two main aspects. First, we demonstrate the appearance of those spin-Floquet states using the ab initio TDDFT scheme[23]. Second, we analyze the obtained Floquet spectra in terms of the second-quantized form of the electron–phonon coupling without resorting to the Ehrenfest semi-classical picture. For the former, considering what we have

learned from the model Hamiltonian, we set up the initial condition of the TDDFT simulation close to the Floquet eigenstate $|\Psi_{\alpha_-}(t=0)\rangle$. To simulate the action of the circularly polarized $E''$ phonon, the initial velocities of the two S atoms are set in the perpendicular direction to the initial atomic displacement. This is based on a classical vibration of a mass constrained by two equal springs placed perpendicularly in a plane, namely $E_{tot} = \frac{1}{2}m(\dot{x}^2+\dot{y}^2)+\frac{1}{2}m\omega^2(x^2+y^2)$. In this case, the vibration of the mass can be polarized in any in-plane direction, for instance $\mathbf{r}(t) = R\sin(\omega t)\hat{x}$ or $\mathbf{r}(t) = R\sin(\omega t)\hat{y}$ and a phase difference between the two degenerate linear motions, namely $x(t) = R\sin(\omega t)$ and $y(t) = R\sin(\omega t - \pi/2)$, results in the circular motion ($E_{tot}=mR^2\omega^2$). We used that the displacement ($R$) and the velocity ($R\omega$) are determined once the total energy and the frequency are defined. The Floquet state spin-trajectory obtained in this ab initio way is presented in Fig. 3d, which confirms the model Larmor-type rotation with a constant $S_z$ value. With increasing amplitude of the $E''$ mode, the precessing Floquet-spin acquires larger components in the $x$–$y$-plane, i.e., we get a smaller $S_z$ value (Supplementary Note 4 and Supplementary Fig. 2).

**SOC effect in terms of phonon quanta.** Moving to the second aspect, instead of having a semi-classical description of phonons, we treat them now in a quantized form by defining the interaction Hamiltonian in terms of electron–phonon coupling with the quantized phonon field[20,29]. Since the valley states of MoS$_2$ have definite angular momentum eigenstates, only one of the circularly polarized phonons can have a non-zero matrix element between the two CBM states[15,29]. In this case, the interaction Hamiltonian including only the zone-center phonon can be written as,

$$\hat{H} = \frac{e}{m}\mathbf{B}\cdot\hat{\mathbf{S}} = \varepsilon_0\hat{\sigma}_z + g\left(\hat{\sigma}_+\hat{b}e^{i\omega_{ph}t}+\hat{\sigma}_-\hat{b}^+e^{-i\omega_{ph}t}\right), \tag{5}$$

where $\hat{b}$ and $\hat{b}^+$ represent annihilation and creation of the right-handed circularly polarized $E''$ phonon. Using $\hat{b}|n\rangle = \sqrt{n}|n-1\rangle$, $\hat{b}^+|n\rangle = \sqrt{n+1}|n+1\rangle$, and assuming a factorized solution $|\psi\rangle\otimes|\text{phonon}\rangle = \sum_n\left(D_{\sigma_1,n}(t)|\sigma_1;n\rangle+D_{\sigma_2,n}(t)|\sigma_2;n\rangle\right)$, where $\sigma_1$ and $\sigma_2$ indicate the spin index of the two CBM bands, the time-dependent Schrödinger equation can be written in terms of these coefficients, as follows (see Supplementary Notes 5 and 6 for a detailed derivation).

$$i\frac{\partial}{\partial t}\begin{pmatrix} D_{\sigma_2,n} \\ D_{\sigma_1,n+1} \end{pmatrix} = \begin{bmatrix} \varepsilon_0 & g\sqrt{n+1}e^{i\omega_{ph}t} \\ g\sqrt{n+1}e^{-i\omega_{ph}t} & -\varepsilon_0 \end{bmatrix}\begin{pmatrix} D_{\sigma_2,n} \\ D_{\sigma_1,n+1} \end{pmatrix}$$
$$= \left[\varepsilon_0\hat{\sigma}_z + g\sqrt{n+1}\left(\cos(\omega_{ph}t)\hat{\sigma}_x -\sin(\omega_{ph}t)\hat{\sigma}_y\right)\right]\begin{pmatrix} D_{\sigma_2,n} \\ D_{\sigma_1,n+1} \end{pmatrix}. \tag{6}$$

This $2\times2$ matrix equation coincides with the aforementioned semi-classical one if we substitute $\varepsilon_{ph} = g\sqrt{n+1}$. This indicates that the Floquet-spin states are quantized in terms of the phonon quantum. For example, the spin Larmor precession of the $\alpha_+$ Floquet state has a constant $S_z$ value of $\langle\Psi_{\alpha_+}(t)|\hat{S}_z|\Psi_{\alpha_+}(t)\rangle = \hbar/\left(2\sqrt{1+g^2(n+1)/\Delta^2}\right)$ (see Fig. 3c and Eq. (4)). A selective pumping of a particularly polarized phonon can be easily obtained if the phonon mode is IR active. It is well known that the $E''$ is not IR active for the monolayer but becomes active for the bilayer[33,34]. This configuration of a bilayer IR active system is achieved in thin films of TMDs[35] or can be constructed through a stacking of van der Waals layers.

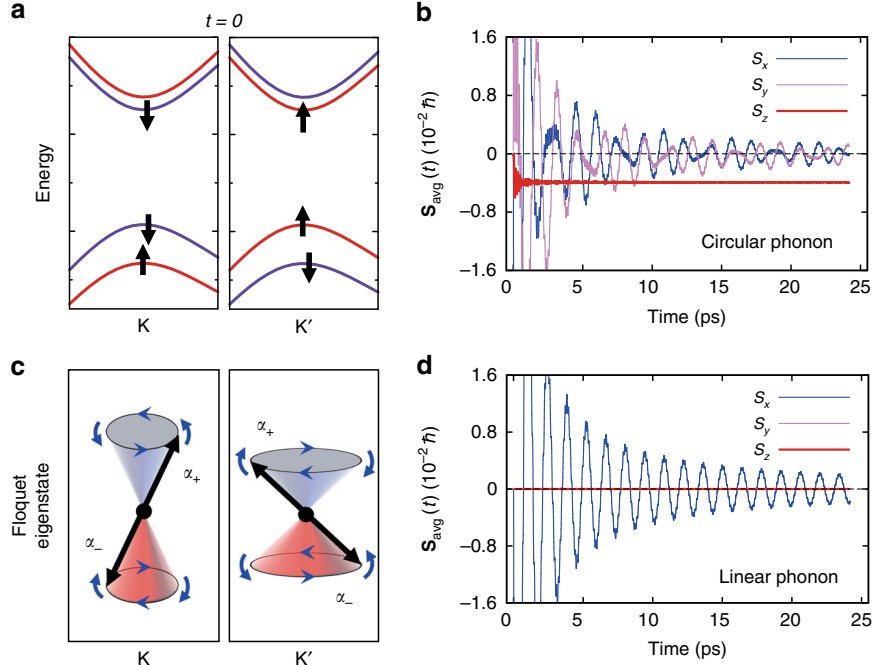

**Fig. 4** Phonon induced spin-Floquet magnetism. **a** Schematic for the electronic configuration with one spin-down and spin-up electron being added onto the K and K' valley, respectively. **b** Cumulative time average of the total spin of $MoS_2$ being driven by the circularly polarized $E''$ phonon. **c** Schematics for the two spin-Floquet eigenstates at K and K' defined by the right-circularly polarized phonon. **d** The same time-averaged total spin with the linearly polarized phonon. Details of each component on each valley are given in Supplementary Fig. 7

**Spin-Floquet valley magnetism**. Finally, we propose that total net spin of the electrons in the valleys can be indeed engineered and controlled through coherent excitation of phonons (in this case of $MoS_2$ by a linear combination of two orthogonal and dephased $E''$ phonons). This can be explicitly discussed in terms of the cumulative time-averaged total spin, defined as $S_{avg}(t) = (1/t) \int_0^t (S_K(\tau) + S_{K'}(\tau)) d\tau$. Suppose that, as an initial configuration, a spin-down and a spin-up electrons are prepared in the CBM edge of the K and K' point, respectively, as depicted in Fig. 4a. The time propagation of the total spin, evolved from this initial configuration, under presence of a right-circularly polarized phonon is presented in Fig. 4b. Remarkably, the time-averaged total spin results in a non-negligible net magnetization even though the system is non-magnetic in its ground state. This surprising result can be explained in terms of the Floquet eigenstates of the driven system. The spinor in the valley can be decomposed into two Floquet states, each one carrying a constant $S_z$ value: $|\psi(t)\rangle_K = D_+ |\Psi_{\alpha_+}(t)\rangle_K + D_- |\Psi_{\alpha_-}(t)\rangle_K$. For the circularly polarized phonon, the Floquet states at K and K' differ from each other, having different $S_z$ values, as schematically depicted in Fig. 4c, leading to the non-zero constant total spin value. Details of the derivation are given in Supplementary Note 7. This behavior is analogous to the observed dichroism for circularly polarized photons[14,15,26]. We note that, in the initial configuration (Fig. 4a), the electron can be paired with its time-reversal symmetric Kramer partner[30], which keeps the system in the non-magnetic state. However, the presence of a circularly polarized phonon makes the system lose time-reversal symmetry, through the discrimination between the two valleys. In contrast, a linearly polarized phonon is not able to distinguish between the two valleys, and the spins started from the initial non-magnetic configuration evolve in time with vanishing average value, as presented in Fig. 4d. We examined the same phonon-driven spin-

Floquet state for the case of TMDC bilayer. For bilayer, the $E''$ phonon separates into two branches, among which only the $E_u$ mode is IR-active and produces the spin-Floquet valley magnetic responses[36,37], as summarized in Supplementary Note 8, Supplementary Table 2, and Supplementary Fig. 5.

The necessary initial condition of this phonon dependent magnetization of the valley electrons can be prepared electrostatically or optically. A positive gating of the $MoS_2$ semiconductor attracts minimal electron carriers onto the CBM edges of the valleys. A selective spin population of the two valleys (up and down electrons at K' and K, respectively) can be achieved in very low temperature, because of the small up/down splitting of the CBM bands. In practice, other members of the 2H-polytype semiconducting TMDCs with a wider SOC splitting can be considered for more efficient electrostatic gating. On the other hand, a narrow-band linearly polarized light in resonance with the band gap can induce the aforementioned electron net spin population in K and K' valleys, which leaves the holes of opposite spin behind. As discussed above, the spins of the holes are almost immobile (up to a few quanta of phonons), and thus the spin dynamics of the system are mainly governed by the motion of CBM electrons. Nevertheless, an experimental realization of the spin-Floquet valley magnetism is quite challenging. In Supplementary Table 3 we provide a full scanning of properties of 2H-polytype semiconducting TMDCs in order to identify the best candidate to exhibit spin-Floquet valley magnetism under realistic experimental conditions. $MoTe_2$ and $WTe_2$ appear as promising candidates due to their large SOC splitting and the phonon frequency near low-IR range.

## Discussion

In summary, by performing full-fledged first principles TDDFT calculations and by analyzing them in terms of a model

Hamiltonian, we found that the lowest-lying optical phonon is delicately coupled to the spin of the electron at the CBM of valleys of MoS$_2$. When a circularly polarized phonon is excited, the spin state at the valley splits into two distinct Floquet states, characterized by Larmor rotations with the amplitude determined by the phonon occupation number. The dichroic response of the circularly polarized phonon makes the pair of valleys lose the time-reversal partnership, and as a result, the electrons in the CBM produce a non-zero out-of-plane magnetization. Our results suggest that advances in polarity-controlled phonon pumping through a coherent laser excitation could be directed to dynamical spin manipulation of a SOC system, which can be developed as a vehicle for quantum computation or spintronics applications.

## Methods

**Computational method and code availability**. The ground state electronic and phonon structure were calculated with the Quantum ESPRESSO package[22]. For the non-collinear Kohn–Sham wavefunctions with SOC interaction, the plane-wave basis set with 30 Ry energy cut-off, the Perdew–Burke–Ernzerhof type gradient approximated exchange-correlation functional[24], and the projector augmented wave method with full-relativistic potential were used[38]. The primitive unit cell with the lattice vector of $a = 3.15$ Å and the vacuum slab of 15 Å vacuum were used to simulate the monolayer MoS$_2$. The whole Bouillon zone was sampled with the grids of $18 \times 18 \times 1$ points excluding any symmetric operation. The Ehrenfest forces were calculated from the instantaneous total energy functional as $\mathbf{F}_\lambda = -\frac{\partial}{\partial \mathbf{R}_\lambda} E_{tot}[\rho(t), \mathbf{R}_\lambda(t)]$[23]. For the computations of the time propagations we used the plane-wave package developed by ourselves[23] and the public-open Octopus package[39]. The package can be released upon request to the authors. We tested the accuracy by varying a few parameters for the time propagation, and the presented results were calculated using the Crank–Nicolson propagator with $\Delta t$=2.42 as, which preserve the total energy within $5.3 \times 10^{-5}$ eV over 245 fs. The reduced $k$-points grid of $6 \times 6 \times 1$ $k$-points were used for the time-evolution which included the occupied VBM states and some of CBM states. More details are given in Supplementary Note 1 and Supplementary Fig. 6.

**Data availability**. The calculated numerical data that support our study are available in "NOMAD repository" with the identifier "https://doi.org/10.17172/NOMAD/2017.11.10-1".

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

## Acknowledgements

We acknowledge the financial support from the European Research Council (ERC-2015-AdG-694097), European Union's H2020 program under GA no. 676580 (NOMAD) and GA no. 646259 (MOSTOPHOS). D.S. and N.P. acknowledge the support from BRL (NRF-2017R1A4A101532).

## Author contributions

D.S. performed the calculation and analyzed the data; N.P., H.H., U.D.G., and A.R. developed the model Hamiltonian and analyzed the solution; D.S., N.P., H.H. A.R., and

U.D.G. edited the first draft of the manuscript. All authors discussed and analyzed the results and contributed and commented on the manuscript.

## Additional information

**Competing interests:** The authors declare no competing financial interests.

