## [Peer Review File · Nature Communications]

Reviewers' comments:

Reviewer #1 (Remarks to the Author):

The manuscript by D. Shin et al. presents an original computational study on Floquet states in quantum matter. The authors focus on the semiconducting MoS₂, which has been studied in the last years as a promising two-dimensional valleytronic material lacking any intrinsic magnetic order. The authors develop a theoretical set-up for triggering the infrared-active E'' phonon out-of-equilibrium and they study the impact of the oscillating mode on the electronic and spin properties of the material. The calculations are performed within the TDDFT formalism and rationalized with minimal model Hamiltonians, providing very robust results. The key observation of this work is the development of a transient out-of-plane magnetization of the conduction band electron spins when the E'' phonon mode is in a coherently-excited circularly polarized state (as resonantly pumped by a circularly polarized laser beam). This result is rather surprising and is extended to the entire class of semiconducting transition-metal dichalcogenides, providing generality to the calculations.

All in all, the set-up developed by the authors is impressive and it represents the state-of-the-art of current computational calculations involving nonequilibrium quantum states of matter. In this respect, the authors go beyond the results obtained in recent papers of similar kind (Nano Lett. 16, 7993 and Nat. Comm. 8, 13940), by including the possibility to excite an infrared-active mode of the system in a circularly-polarized state and monitoring the spin response of the system. As such, I recommend the publication of this manuscript in Nature Communications, as the presented research is of high quality. Moreover, the topic of Floquet states is very hot at the moment and this work will stimulate deep discussion in the field.

However, prior to publications, the authors should deal with some modifications of their current manuscript. I express here below my requests:

1) The authors should be aware that the experimental realization of their theoretical prediction is very challenging (probably impossible at the moment), due to constraints in current laser technology. In this respect, the authors are silent, just discussing how the electron population can be prepared in the conduction band. Within the all-optical realization, this experiment should take the form of a pump-pump-probe, with the first pump (circularly polarized) promoting a selective particle-hole excitation in the K or K' valleys, the second pump driving the desired phonon mode out-of-equilibrium (resonant excitation) and the probe pulse monitoring the spin state of the system. Therefore, the authors should add a more compelling "applied" discussion in which they present the experimental feasibility of such an experiment, including advantages and drawbacks of the different ultrafast techniques that can be employed (spin-resolved pump-probe ARPES, Faraday rotation, etc.). This would be an important step for guiding experimentalists in realizing this predicted state of matter and/or to aliment the discussion on alternative pathways to realize Floquet states. This discussion would also enrich the paper giving more value to the current results.

2) Following point 1), another important constraint from an experimental point of view is the fact that the system is a monolayer two-dimensional material. This makes the experimental observation of these effects rather unreachable due to constraints in crystal fabrication technology and sensitivity of pump-probe experiments. It would be a breakthrough if the current results could be extended to the case of quasi-2D systems, such as the bilayer MoS₂ or its bulk form. In such a case, the electronic and lattice structure is expected to change dramatically, but the pump-probe experiments would be performed close to the surface of these materials. Could the authors just add some sentences on what they expect in the bilayer or bulk limit?

3) From a theoretical perspective, the calculations have been performed with TDDFT, thus neglecting the presence of excitonic effects. Excitonic effects are well known to be present in transition metal dichalcogenides such as MoS₂. I understand that going to this level of theory

would require the solution of the Bethe-Salpeter equation under nonequilibrium conditions and this is a very challenging task at the moment. However, it would be important that the authors stress this point and try to envision the presence of excitonic couplings in the all-optical pump-probe scenario. To my understanding, this would probably imply the creation of a new quasiparticle that hybridizes excitons, phonons and spin degrees of freedom.

Minor points:

4) I am not sure that the word "semiconducting transition-metal dichalcogenides" is a very precise one to refer to the generality of the calculations. The reason is that part of the community refers also to transition-metal dichalcogenides with charge-density wave instabilities as semiconducting. Here, the semiconducting behavior (e.g. the low-energy gap) rises as a result of an instability in a metallic system (see TaS₂). In this case, I don't expect the results envisioned by the authors to be at play. To avoid any confusion, it would be important to reformulate the generality criterion for these calculations, specifying the origin of the semiconducting gap (no charge-density waves, no Mott correlations).

5) One aspect that I got confused from at the beginning regards the polarization of the excited phonon mode(s) in Fig. 1. Indeed, in the abstract, the authors state that the E'' phonon has a circular polarization. It would be important that the authors specify more clearly in their first calculations that the polarization of the excited phonon is linear.

6) Typos: "Low terahertz frequency" "Low-frequency terahertz"
"That can have with very distinct" "That can have very distinct"

Reviewer #2 (Remarks to the Author):

This paper studies which is the effect of different phonons coupled to the spins in MoS₂ by means of TDDFT calculations.

They show that the only phonon that couples to the spin is the E'' while the others do not affect the spins at all. They also find that, when the E'' phonon has a circular polarization, a net magnetization appears in the system.

Finally, an effective Hamiltonian that reproduces qualitatively the TDDFT results is developed which would be useful in order to make longer calculations.

The paper is well written and presents state-of-the-art TDDFT calculations and could be of interest for the community.

In the following, I present some minor remarks and questions for the authors regarding the article that, in my opinion, should be clarified before publishing the manuscript:

1) It is shown by the calculations that the only phonon that couples to the spin is the E'' phonon. Is there any physical argument behind this fact? Why do the other phonons do not couple?

2) In page 5, it is stated that the spins in the VBMs remain in the equilibrium configuration during the time evolution due to the fact that their up/down splitting is an order of magnitude larger than the phonon energy quantum (see extended data Fig. 2). Looking at the extended data this point is not clear for me. Could the authors comment and clarify this?

3) In page 14, the authors claim that the E'' phonon is not IR active for the monolayer but becomes active for the bilayer. I have several questions regarding this:

3.1) If I understood correctly, the only way to make the system non-magnetic is to pump circularly polarized phonons to the system. Since this is one of the main highlights of the paper, do

the authors have any proposal for the pumping of this phonons?

3.2) Although in the bilayer is active, in this case, the VBM and CBM are not anymore in the K and K' points, but in the Γ and a k-point between K and M which is not a special one, respectively. Would all the arguments that are made for the monolayer case still hold for the bilayer case?

4) I think there is an error in extended data table 1 (second row, third column): the atomic displacement should be in the y-direction.

Reviewers' comments:

Reviewer #1 (Remarks to the Author):

The manuscript by D. Shin et al. presents an original computational study on Floquet states in quantum matter. The authors focus on the semiconducting MoS₂, which has been studied in the last years as a promising two-dimensional valleytronic material lacking any intrinsic magnetic order. The authors develop a theoretical set-up for triggering the infrared-active E'' phonon out-of-equilibrium and they study the impact of the oscillating mode on the electronic and spin properties of the material. The calculations are performed within the TDDFT formalism and rationalized with minimal model Hamiltonians, providing very robust results. The key observation of this work is the development of a transient out-of-plane magnetization of the conduction band electron spins when the E'' phonon mode is in a coherently-excited circularly polarized state (as resonantly pumped by a circularly polarized laser beam). This result is rather surprising and is extended to the entire class of semiconducting transition-metal dichalcogenides, providing generality to the calculations.

All in all, the set-up developed by the authors is impressive and it represents the state-of-the-art of current computational calculations involving nonequilibrium quantum states of matter. In this respect, the authors go beyond the results obtained in recent papers of similar kind (Nano Lett. 16, 7993 and Nat. Comm. 8, 13940), by including the possibility to excite an infrared-active mode of the system in a circularly-polarized state and monitoring the spin response of the system. As such, I recommend the publication of this manuscript in Nature Communications, as the presented research is of high quality. Moreover, the topic of Floquet states is very hot at the moment and this work will stimulate deep discussion in the field.

However, prior to publications, the authors should deal with some modifications of their current manuscript. I express here below my requests:

[Question 1]

1) The authors should be aware that the experimental realization of their theoretical prediction is very challenging (probably impossible at the moment), due to constraints in current laser technology. In this respect, the authors are silent, just discussing how the electron population can be prepared in the conduction band. Within the all-optical realization, this experiment should take the form of a pump-pump-probe, with the first pump (circularly polarized) promoting a selective particle-hole excitation in the K or K' valleys, the second pump driving the desired phonon mode out-of-equilibrium (resonant excitation) and the probe pulse monitoring the spin state of the system. Therefore, the authors should add a more compelling "applied" discussion in which they present the

experimental feasibility of such an experiment, including advantages and drawbacks of the different ultrafast techniques that can be employed (spin-resolved pump-probe ARPES, Faraday rotation, etc.). This would be an important step for guiding experimentalists in realizing this predicted state of matter and/or to aliment the discussion on alternative pathways to realize Floquet states. This discussion would also enrich the paper giving more value to the current results.

[Our answer]

This reviewer's comments are very relevant and helpful. Beyond the results presented in the first submitted version, we extended our calculation over to the whole family of TMDC. Meanwhile, we had a few private communications with experimental groups to understand the prerequisites for the experiment. To realize the spin-Floquet valley magnetism in an actual experiment, a spin-resolved population of electrons in the valleys of CBM is required to provide a time-reversal symmetric initial electronic configuration. This can be prepared either through electron-hole excitations at the valleys or electrostatic electron doping. To achieve it in the former way, as pointed out by the reviewer, a sophisticated pump-pump-probe type of experiment is necessary. In these processes, we think that various 2H-phase TMDCs are almost equally preferable.

On the other hand, the electrostatic electron doping can provide a simpler route to the desired initial condition. We examined the size of SOC splitting of the family of 2H-polytype TMDC. The results are summarized in the Supplementary Table 2 in the revised version of SI. This part of SI is cited in the page 14 of the revised main text. The SOC splitting of the CBM valleys for MoS₂ is rather small, while some others such as MoTe₂ and WTe₂ have quite appreciable SOC splitting. Based on this result, we see that the CBM valleys of MoTe₂ and WTe₂ can accommodate the spin-resolved electron population at nitrogen temperature or even near room temperature. In this regard of the electrostatic preparation, MoTe₂, WTe₂ and WSe₂ would be a better candidate than MoS₂. We verified that the CBM spins of those TMDCs (MoTe₂, WTe₂ and WSe₂) exhibit similar flexibility as that of MoS₂ in response to the lattice displacement along the E'' phonon eigenvectors. The spin dynamics for the case of WTe₂ are presented in Supplementary Figure 4 in the revised version of SI.

On the other hand, in terms of currently affordable laser condition, the IR-active phonon pumping is more easily achieved for the frequencies below or around 3 THz or above 15THz. Taking this factor as well into account, we see that MoTe₂ (3.3 THz) or WTe₂ (3.4 THz) could be an even better candidate. We agree with the reviewer and concede that the experimental realization is challenging. Nevertheless, based on our private communications, we foresee different groups trying to realize it after this work is published.

[Revised part in the main text: page 13, line 274]

In practice, other members of the 2H-polytype semiconducting TMDCs with a wider SOC splitting can be considered for more efficient electrostatic gating.

[Revised part in the main text: page 13, line 280]

Nevertheless, an experimental realization of the spin-Floquet valley magnetism is quite challenging. In Supplementary Table 2 we provide a full scanning of properties of 2H-polytype semiconducting TMDCs in order to identify the best candidate to exhibit spin-Floquet valley magnetism under realistic experimental conditions. MoTe₂ and WTe₂ appear as promising candidates due to their large SOC splitting and the phonon frequency near low-IR range.

[Updated part in SI]

TMDCs	$\Delta E_{\text{VBM, spin-split}}$	$\Delta E_{\text{CBM, spin-split}}$	E'' phonon
CrS ₂	70 meV	3.6 meV	7.8 THz
CrSe ₂	92 meV	16 meV	4.4 THz
CrTe ₂	109 meV	21 meV	3.1 THz
MoS ₂	150 meV	3 meV	8.2 THz
MoSe ₂	188 meV	20 meV	4.8 THz
MoTe ₂	219 meV	32 meV	3.3 THz
WS ₂	440 meV	28 meV	8.5 THz
WSe ₂	481 meV	34 meV	4.9 THz
WTe ₂	504 meV	48 meV	3.4 THz

Supplementary Table 2 | The SOC splitting at VBM and CBM, and the E'' phonon frequencies of various TMDCs. To achieve a spin-resolved electron population in CBM valleys through the electrostatic doping, a larger SOC splitting is favored, and thus MoTe₂, WTe₂ and WSe₂ are better candidates for a realistic experiment.

Supplementary Figure 4 | The time evolution profile of the CBM spins at K valley of the monolayer WTe₂ on the presence of a linearly polarized E'' phonon. a-b, The initial state was prepared through (a) the electron-hole excitation and (b) the electrostatic electron doping. The left panels in (a) and (b) depicts the initial electronic configuration. The hole spins in (a) remained almost rigid throughout the simulation time, and the presence of hole in the VBM has negligible effect on the dynamics of the CBM spin. In this simulation, the phonon energy is fitted to the zero-point vibration, that is, $E_{\text{ph}} = \frac{1}{2} \hbar \omega_{\text{ph}}$ which correspond to $\epsilon_{\text{ph}} = 0.44 \epsilon_0$, where ϵ_0 is the CBM splitting. The same calculation with single phonon ($E_{\text{ph}} = \frac{3}{2} \hbar \omega_{\text{ph}}$) led to similar result.

[Question 2]

2) Following point 1), another important constraint from an experimental point of view is the fact that the system is a monolayer two-dimensional material. This makes the experimental observation of these effects rather unreachable due to constraints in crystal fabrication technology and sensitivity of pump-probe experiments. It would be a breakthrough if the current results could be extended to the case of quasi-2D systems, such as the bilayer MoS₂ or its bulk form. In such a case, the electronic and lattice structure is expected to change dramatically, but the pump-probe experiments would be performed close to the surface of these materials. Could the authors just add some sentences on what they expect in the bilayer or bulk limit?

[Our Answer]

The reviewer's comments are very relevant. We extended our investigation to the TMDC bilayers. A distinct character of the bilayer is the emergence of the indirect nature in the band gap structure. A thicker layer or the surface layers of the bulk sample can have similar features. For the bilayer, the VBM locates obviously on the Γ as a result of interlayer hybridization. On the other hand, the structure of CBM has some complexity. Whether the energy minimum is on the K(K') or the Σ point depends on stacking configuration and also computational method. For example, a recent GGA+GW calculation (Phys. Rev. B **89**, 205311 (2014)) reported that all 2H-polytype TMDC bilayers have its CBM minimum on the Σ point.

The same literature, on the other hand, showed that heterogeneous layers (such as MoS₂/MoSe₂, MoSe₂/MoTe₂, WSe₂/WS₂, WSe₂/WTe₂) have their CBM on K(K'). Another previous study (Nanoscale **6**, 4566 (2014)) reported that, as a result of the calculation using HSE06 hybrid functional, heterogeneous trilayers have their CBMs on K(K').

The key underlying mechanism for the spin-Floquet valley magnetism is the coupling between the E'' phonon and the spinor at the K and K' valleys of the CBM. In this regard, the model for the spin-Floquet state, we presented in the main text for the case of the monolayer, can be applied to the case of the bilayers when its CBM sits on K and K'. Even when the CBM locates on the Σ point, our model for the spin-Floquet valley magnetism is still valid because the spins on the CBM of the Σ is as rigid as the VBM of K. We would like to emphasize that, when electrons are populated in minima of CBM, only the spin moments in K and K' reacts to the presence of E'' phonon. To prove this, we calculated the variation of the spin direction of the CBM of K, CBM of Σ , and VBM of K, as summarized in the Supplementary Figure 6 in the revised SI.

We would like to attract reader's attention on the increased number of valleys. For example in the case of bilayer, there are four valleys at each K and K' on up and down layers. With this increased number of valleys, the magnitude of the dichroic spin-Floquet valley-magnetism, with a given circular phonon, additively increases compared with that of the monolayer. The calculated effective Hamiltonian and the resulting cumulative time-averaged spin values are summarized in the Supplementary Table. 3 and in the Supplementary Figure 5 in the revised SI.

[Revised part in the main text: page 13, line 264]

We examined the same phonon-driven spin-Floquet state for the case of TMDC bilayer. For bilayer, the E'' phonon separates into two branches, among which only the E_u mode is IR-active and produces the spin-Floquet valley magnetic responses, as summarized in Supplementary Discussion 8, the Supplementary Table 3, and in the Supplementary Fig. 5 in the SI^{36,37}.

[Revised part of SI]

a

E_u Phonon		E_g Phonon	
8.22 THz		8.26 THz	
IR-active		Raman-active	

b

E_u phonon mode	K	K'
Upper layer 	$\vec{S}_K^U(t=0) = \frac{\hbar}{2} \hat{z}$ $\hat{H} = \begin{pmatrix} -\epsilon_0 & -\epsilon_{ph} e^{i\omega_{ph}t} \\ -\epsilon_{ph} e^{-i\omega_{ph}t} & \epsilon_0 \end{pmatrix}$ $\Rightarrow \Delta_U = \frac{\omega_{ph}}{2} - \epsilon_0$	$\vec{S}_{K'}^U(t=0) = -\frac{\hbar}{2} \hat{z}$ $\hat{H} = \begin{pmatrix} \epsilon_0 & \epsilon_{ph} e^{i\omega_{ph}t} \\ \epsilon_{ph} e^{-i\omega_{ph}t} & -\epsilon_0 \end{pmatrix}$ $\Rightarrow \Delta_{K'} = \frac{\omega_{ph}}{2} + \epsilon_0$
Lower layer 	$\vec{S}_K^L(t=0) = -\frac{\hbar}{2} \hat{z}$ $\hat{H} = \begin{pmatrix} \epsilon_0 & -\epsilon_{ph} e^{i\omega_{ph}t} \\ -\epsilon_{ph} e^{-i\omega_{ph}t} & -\epsilon_0 \end{pmatrix}$ $\Rightarrow \Delta_{K'} = \frac{\omega_{ph}}{2} + \epsilon_0$	$\vec{S}_{K'}^L(t=0) = \frac{\hbar}{2} \hat{z}$ $\hat{H} = \begin{pmatrix} -\epsilon_0 & \epsilon_{ph} e^{i\omega_{ph}t} \\ \epsilon_{ph} e^{-i\omega_{ph}t} & \epsilon_0 \end{pmatrix}$ $\Rightarrow \Delta_U = \frac{\omega_{ph}}{2} - \epsilon_0$

Supplementary Table 3 | Phonon-driven spin-Floquet magneto-valleytronics in TMDC

bilayer. a, The phonon eigenvectors, corresponding to the E'' in the case of monolayer, have two branches in bilayer: E_u and E_g . **b**, The model Hamiltonian for the dichroic behavior of the spin-Floquet states of valleys with respect to a circularly polarized E_u phonon. Note that the spin-Floquet state can be described by the same form as eq. 4 in the main text, once the Δ values are defined, as given in this table.

Supplementary Figure 5 | The time-averaged total spin values of the monolayer and bilayer WTe₂ with an E'' and an E_u phonon mode, respectively. a-b, The cumulative time average of the total spin in monolayer WTe₂ with (a) linear and (b) circular polarized E'' phonon. **c-d,** The same total spin of the bilayer WTe₂ with a circular polarized E_u phonon energy with (c) zero-point phonon ($\frac{1}{2}\hbar\omega_{\text{ph}}$) and (d) single phonon ($\frac{3}{2}\hbar\omega_{\text{ph}}$).

Supplementary Figure 6 | Electronic and spin configuration of monolayer MoS₂ with and without a static lattice displacement along an E'' phonon eigenvector. a, Band structure of the equilibrium monolayer MoS₂. b-d, The variations in the spin angle and the SOC splitting ($\Delta\varepsilon$) with respect to the magnitude of the displacement at (b) CBM of Σ point, (c) CBM of K point, and (d) VBM of K point.

Supplementary Discussion 8. Spin-Floquet magneto-valleytronics in bilayers of TMDC.

We extended the model Hamiltonian study of spin-Floquet magneto-valleytronics to the cases of TMDC bilayers. The E'' phonons in the bilayer have two branches, E_u and E_g , which are depicted in Supplementary Table 3a. Details of the bilayer phonon structures can be found in a recent literature.^{S1} We focus on the coupling of the CBM spinors to E_u phonon

which is known to be IR-active. There are four valleys in the case of the bilayer: K and K' in each of upper and lower layers. The spin on the K point of both the upper and lower layer experience the same in-plane magnetic field, in response to a E_u phonon, while the out of plane magnetic fields of them are directed oppositely. As explained in the case of monolayer, the time-reversal partners (the spin on K' point) in each layer are subjected to exactly opposite direction of in-plane and out-of-plane magnetic field. As a result, the spin-up states (that of K in the upper layer and that of K' in lower layer) evolves with the left-handed Hamiltonian, whereas the spin-down states (that of K' in the upper layer and K in the lower layer) experiences the right-handed Hamiltonian, as given in Supplementary Table.3. The dichroic behavior of the circularly polarized E_u phonon can be parametrized by the Δ parameters. For the same phonon with the opposite circular polarity, the parameters for the four valleys need to be interchanged: Δ_R to Δ_L and vice versa. The same study revealed that the E_g phonon drives the spin in the upper layer exactly opposite to those in the lower layer, keeping the time-reversal symmetry at all time. However, this uninteresting E_g mode is not IR-active, and thus a real experiment can excite only E_u avoiding E_g .

The time-averaged total spin of the bilayer WTe_2 is summarized in Supplementary Fig. 5. As a result of the interplay between the opposite spin direction between the layers and the opposite effective magnetic field between the time-reversal partners, the bilayer produces almost twice increased S_z value while the noisy in-plane component is cancelled in the time-averaged profile, as shown in Supplementary Fig. 5c and 5d.

Here, we would also like to summarize the features of the electronic structure of the bilayer. A distinct character of the bilayer (or a thicker layer) is the emergence of the indirect nature in the band gap. The VBM locates obviously on the Γ -point as a result of interlayer hybridization. On the other hand, the structure of CBM of the bilayer is not so obvious.

Whether the energy minimum is on $K(K')$ or Σ depends on stacking configuration and also computational method. For example, a recent GGA+GW calculation reported that all 2H-polytype TMDC bilayers have its CBM minimum on Σ .^{S2} In the same literature, on the other hand, it was shown that heterogeneous layers (such as $\text{MoS}_2/\text{MoSe}_2$, $\text{MoSe}_2/\text{MoTe}_2$, WSe_2/WS_2 , $\text{WSe}_2/\text{WTe}_2$) have their CBM on $K(K')$. Another previous study using HSE06 hybrid functional reported that heterogeneous trilayers have their CBMs on $K(K')$.^{S3} Thus, there are numerous combinations of the bilayer on which the phonon driven spin-Floquet magnetism can be realized. Even when the CBM locates on Σ , our model for the spin-Floquet valley magnetism is still valid because the spins on the CBM of Σ is as rigid as the VBM of K . To prove this, we calculated the variation of the spin direction of the CBM of K , CBM of Σ , and VBM of K , as summarized in the Supplementary Fig. 6. It shows that the spins at CBM of Σ and VBM of K remain near the equilibrium direction irrespective of the lattice displacement.

[Question 3]

3) From a theoretical perspective, the calculations have been performed with TDDFT, thus neglecting the presence of excitonic effects. Excitonic effects are well known to be present in transition metal dichalcogenides such as MoS_2 . I understand that going to this level of theory would require the solution of the Bethe-Salpeter equation under nonequilibrium conditions and this is a very challenging task at the moment. However, it would be important that the authors stress this point and try to envision the presence of excitonic couplings in the all-optical pump-probe scenario. To my understanding, this would probably imply the creation of a new quasiparticle that hybridizes excitons, phonons and spin degrees of freedom.

[Our answer]

In the revised SI, the Supplementary Figure 4 shows that the dynamics of the CBM valley spin of the monolayer WTe_2 is negligibly affected by the presence of a hole. During this simulation window, the hole spin remained almost rigid. This indicates that the spin-Floquet valley magnetic state is mainly determined by the electrons on the CBM valley, and the presence of holes, even with the excitonic binding, is not expected to have a substantial

effect. Excitonic effects in TMDC valleys were recently studied by several authors, and it is well accepted that the optically pumped excitonic valley state decays in few ps by inter-valley scattering. We conjecture that the phonon-coupled CBM spinor at the valleys, which has a femtosecond movement in its spin direction, can possibly affect the exciton life time owing to the varied spin direction. However, we think that such excitonic effect is beyond the scope of the present study. Moreover, although many body perturbations (such as Bethe-Salpeter) or TDDFT have been used to describe such two-body excitation of materials, we must concede that dealing with them in ab initio real-time scheme is far beyond the current capability.

Minor points:

[Question 4]

4) I am not sure that the word “semiconducting transition-metal dichalcogenides” is a very precise one to refer to the generality of the calculations. The reason is that part of the community refers also to transition-metal dichalcogenides with charge-density wave instabilities as semiconducting. Here, the semiconducting behavior (e.g. the low-energy gap) rises as a result of an instability in a metallic system (see TaS₂). In this case, I don’t expect the results envisioned by the authors to be at play. To avoid any confusion, it would be important to reformulate the generality criterion for these calculations, specifying the origin of the semiconducting gap (no charge-density waves, no Mott correlations).

[Our answer]

We do wish to attract a general readership, and we agree that any part of the community should accept our result without any confusion. As already pointed by the reviewer, here, we don’t deal with any of Fermi-instability nature of the metallic TMDC (One notable example is PRL 112, 157601(2014)). The band structure we are dealing with in this study is the 2H-polytype TMDCs whose band gaps are larger than 1 eV. Regarding the specific naming issue, we have a slightly different view from the reviewer. We would rather see that these types of 2H-shape TMDCs are commonly referred to as “semiconducting ones”, which describes the obvious band gap in the order of 1 eV. Nevertheless, we understand the reviewer’s concern, and, in this regard, we included the explicit expression of “2H semiconducting transition metal dichalcogenides” in abstract and in the part of the introduction, to clearly distinguish it from the metallic phases.

[Revised part in the abstract: page 1, line 20]

This dichroic magnetic response of the valley states is general for all 2H semiconducting transition-metal dichalcogenides and can be probed and controlled by infrared coherent laser excitation.

[Question 5]

5) One aspect that I got confused from at the beginning regards the polarization of the excited phonon mode(s) in Fig. 1. Indeed, in the abstract, the authors state that the E'' phonon has a circular polarization. It would be important that the authors specify more clearly in their first calculations that the polarization of the excited phonon is linear.

[Our answer]

We appreciate much for reviewer's detailed reading and careful consideration of the readability. Accordingly, we stated explicitly the phonon polarity in the revised caption of Fig. 1.

[Revised part in the Figure caption: page 18, line 409]

Figure 1 | TDDFT simulation of the spin-phonon dynamics of monolayer MoS₂. **a**, The initial electronic configuration used here with one electron excited at the K valley, for example, by right-handed light. **b**, Top view of the solid and a side view of the eigenvectors of the four zone-center optical phonons. **c**, The time trace of the spin vector of the CBM electron with the E'' phonon being coherently excited. **d,e,f**, The same time profiles of the Cartesian components of the spin with each of the phonon modes E'' , E' , A_1 , A_2'' and with the frozen lattice. The full phonon dispersion is given in Supplementary Fig. 1. Here, all phonons are linearly polarized in y-direction.

[Question 6]

6) Typos: “Low terahertz frequency” “Low-frequency terahertz”
“That can have with very distinct” “That can have very distinct”

[Our answer]

We are grateful for reviewer’s careful reading. We modified it accordingly and examined the other part of the text.

[Revised part in the main text: page 2, line 28]

Low-frequency terahertz sources are expected to have particular relevance here as they allow for a straightforward control avoiding high energy transfer to the material⁹.

[Revised part in the main text: page 2, line 34]

We note that all semiconducting two-dimensional (2D) materials have time-reversal symmetry partners (or valleys) with very distinct electronic and spin properties.

Reviewer #2 (Remarks to the Author):

This paper studies which is the effect of different phonons coupled to the spins in MoS₂ by means of TDDFT calculations.

They show that the only phonon that couples to the spin is the E’ while the others do not affect the spins at all. They also find that, when the E’ phonon has a circular polarization, a net magnetization appears in the system.

Finally, an effective Hamiltonian that reproduces qualitatively the TDDFT results is developed which would be useful in order to make longer calculations.

The paper is well written and presents state-of-the-art TDDFT calculations and could be of interest for the community.

In the following, I present some minor remarks and questions for the authors regarding the article that, in my opinion, should be clarified before publishing the manuscript:

[Question 1]

1) It is shown by the calculations that the only phonon that couples to the spin is the E’ phonon. Is there any physical argument behind this fact? Why do the other phonons do not

couple?

[Our answer]

All the SOC effects of the phonon came through the gradient of the scalar potential, as given by the operator form of $\hat{V}_{SOC} = -\frac{\hbar q}{4m^2c^2} \hat{\sigma} \cdot (\nabla V \times \hat{p})$. This operator without the operand does not produce any quantitative result. Instead, we summarized the overall result of the electronic structure, as a result of the SOC effect of the lattice motion, in Figure 2 of the main text. A qualitative interpretation of these results can be derived from the symmetry of the lattice and phonon vector. In the equilibrium structure, the TMDC layer has a mirror plane, and the spin is restricted to the out of plane direction. Since the emergence of the mirror-symmetry breaking phonons releases this constraint, a deflection of the spin can be allowed. Among the optical phonons, only E'' and A_2'' modes vibrate against the mirror symmetry, while others vibrate within the mirror symmetry. The difference between E'' and A_2'' can possibly be attributed to the in-plane rotational symmetry. As presented in the Supplementary Figure 7, the CBM is derived from d_{z^2} orbital. While E'' breaks the in-plane trigonal symmetry, the vibration in A_2'' preserve the in-plane rotation symmetry, which share the same feature as the d_{z^2} orbital. However, instead of this qualitative understanding, we see that the results summarized in Figure 2 indicate the effect of SOC more evidently, providing a solid basis for the ensued model Hamiltonian study. Even though we did not spend much part of the text in this qualitative understanding of the SOC, we added some remarks of the symmetry features of the phonon eigenvector.

[Revised part in the main text: page 4, line 78]

We highlight that only the E'' optical phonon mode appreciably couples to the spin motion, while the other three phonons (E' , A_1 , and A_2'') are basically uncoupled. This particular feature can be related to the fact that the E'' is the only optical vibration that breaks both the mirror and trigonal symmetries of the MoS_2 plane, while others preserve at least one of the symmetries.

[Revised part of SI]

Supplementary Figure 7 | Dominant orbital character in VBM and CBM states. a-b,

Orbital character of (a) VBM and (b) CBM valleys of monolayer MoS₂.

[Question 2]

2) In page 5, it is stated that the spins in the VBMs remain in the equilibrium configuration during the time evolution due to the fact that their up/down splitting is an order of magnitude larger than the phonon energy quantum (see extended data (supplementary) Fig. 2). Looking at the extended data (supplementary information) this point is not clear for me. Could the authors comment and clarify this?

[Our answer]

In page 5 of the main text, we meant to state that the phonon energy quantum (a few tens meV, as shown in the Supplementary Figure 2d) is an order of magnitude smaller than that of SOC splitting in VBM, which is 150 meV for the case of MoS₂. This energy scale obviously indicates that the spin state dressed by a few phonons can hardly deflect the spin from the out of plane orientation. Note that the electron-phonon coupling strength (eq. 6 in the main text) increases only by the square root of the phonon number. To prove this feature more evidently, we investigated the variation of the spin inclination angle by a static atomic displacement along an E'' eigenvector, as shown in Supplementary Figure 6. These results obviously indicate that spin direction of the VBM state is negligibly affected by the motion

of the phonon. We modified parts of text to remove the possible confusion raised by the reviewer.

[Revised part in the main text: page 5, line 91]

Moreover, the spins in the VBMs mostly remain near the equilibrium configuration during the time evolution, which can be inferred from the fact that their intrinsic up/down spin splitting is an order of magnitude larger than the phonon induced in-plane SOC magnetic field (given in the Supplementary Figs. 2d and 6d).

[Revised part of SI]

Supplementary Figure 6 | Electronic and spin configuration of monolayer MoS₂ with and without a static lattice displacement along an E'' phonon eigenvector. a, Band structure of the equilibrium monolayer MoS₂. **b-d,** The variations in the spin angle and the

SOC splitting ($\Delta\mathcal{E}$) with respect to the magnitude of the displacement at **(b)** CBM of Σ point, **(c)** CBM of K point, and **(d)** VBM of K point.

[Question 3]

3) In page 14, the authors claim that the E'' phonon is not IR active for the monolayer but becomes active for the bilayer. I have several questions regarding this:

3.1) If I understood correctly, the only way to make the system non-magnetic is to pump circularly polarized phonons to the system. Since this is one of the main highlights of the paper, do the authors have any proposal for the pumping of this phonons?

3.2) Although in the bilayer is active, in this case, the VBM and CBM are not anymore in the K and K' points, but in the Γ and a k-point between K and M which is not a special one, respectively. Would all the arguments that are made for the monolayer case still hold for the bilayer case?

[Our Answer]

We would like to respond first to reviewer's question (3.2). A distinct character of the bilayer (or a thicker layer) is the emergence of the indirect nature in the band gap. The VBM locates obviously on the Γ -point as a result of interlayer hybridization. On the other hand, the structure of CBM of the bilayer is not so obvious. Whether the energy minimum is on K(K') or Σ depends on stacking configuration and also computational method. For example, a recent GGA+GW calculation (Phys. Rev. B **89**, 205311 (2014)) reported that all 2H-polytype TMDC bilayers have its CBM minimum on the Σ point. In the same literature, on the other hand, it was shown that heterogeneous layers (MoS₂/MoSe₂, MoSe₂/MoTe₂, WSe₂/WS₂, WSe₂/WTe₂) have their CBM on K(K'). Another previous study using HSE06 hybrid functional (Nanoscale **6**, 4566 (2014)) reported that heterogeneous trilayers have their CBMs on K(K').

To realize the spin-Floquet magneto-valleytronics in actual experiment, electrons need to be populated in the valleys of CBM of TMDC, as an initial state possessing the time-reversal symmetry. Our study is not much concerned with the VBM structures. Considering aforementioned theoretical results, there are a number of combinations of homo and hetero bilayer on which the phonon-driven spin-Floquet magnetism can be realized. As summarized in the Supplementary Table 3, the model Hamiltonian formalism, that we used

for the spin-Floquet state of the monolayer, can be valid for the bilayer when its CBM sits on K and K'. Even when the CBM locates on Σ -point, our model for the spin-Floquet valley magnetism is still valid, because the spins on the CBM of Σ -point is as rigid as that of the VBM of K. To prove this, we calculated the variation of the spin direction of the CBM of K, CBM of Σ , and VBM of K, as summarized in the Supplementary Figure 6 in the revised SI.

In response to the question on the IR-activity, we would like to point out that the E'' phonon, in bilayers, separates into two branches, E_u and E_g , which are depicted in Supplementary Table 3(a). Details of the bilayer phonon structures can be found in Nano Letters **13**, 1007 (2013). The model Hamiltonian study of the spin-Floquet valley magnetism, with respect to E_u phonon, is summarized in Supplementary Table 3, Supplementary Figure 5, and the Supplementary Discussion 8 in the revised SI. It is noteworthy that the dichroic spin-Floquet valley-magnetism of each layer adds up constructively with a given circular E_u phonon, while those of each layer destructively cancel each other when a circular E_g phonon was turned on. Interestingly, only E_u phonon is IR-active.

[Revised part in the main text: page 13, line 264]

We examined the same phonon-driven spin-Floquet state for the case of TMDC bilayer. For bilayer, the E'' phonon separates into two branches, among which only the E_u mode is IR-active and produces the spin-Floquet valley magnetic responses, as summarized in Supplementary Discussion 8, the Supplementary Table 3, and in the Supplementary Fig. 5 in the SI^{36,37}.

a

E_u Phonon		E_g Phonon	
8.22 THz		8.26 THz	
IR-active		Raman-active	

b

E_u phonon mode	K	K'
Upper layer 	$\vec{S}_K^U(t=0) = \frac{\hbar}{2} \hat{z}$ $\hat{H} = \begin{pmatrix} -\epsilon_0 & -\epsilon_{ph} e^{i\omega_{ph}t} \\ -\epsilon_{ph} e^{-i\omega_{ph}t} & \epsilon_0 \end{pmatrix}$ $\Rightarrow \Delta_L = \frac{\omega_{ph}}{2} - \epsilon_0$	$\vec{S}_{K'}^U(t=0) = -\frac{\hbar}{2} \hat{z}$ $\hat{H} = \begin{pmatrix} \epsilon_0 & \epsilon_{ph} e^{i\omega_{ph}t} \\ \epsilon_{ph} e^{-i\omega_{ph}t} & -\epsilon_0 \end{pmatrix}$ $\Rightarrow \Delta_R = \frac{\omega_{ph}}{2} + \epsilon_0$
Lower layer 	$\vec{S}_K^L(t=0) = -\frac{\hbar}{2} \hat{z}$ $\hat{H} = \begin{pmatrix} \epsilon_0 & -\epsilon_{ph} e^{i\omega_{ph}t} \\ -\epsilon_{ph} e^{-i\omega_{ph}t} & -\epsilon_0 \end{pmatrix}$ $\Rightarrow \Delta_R = \frac{\omega_{ph}}{2} + \epsilon_0$	$\vec{S}_{K'}^L(t=0) = \frac{\hbar}{2} \hat{z}$ $\hat{H} = \begin{pmatrix} -\epsilon_0 & \epsilon_{ph} e^{i\omega_{ph}t} \\ \epsilon_{ph} e^{-i\omega_{ph}t} & \epsilon_0 \end{pmatrix}$ $\Rightarrow \Delta_L = \frac{\omega_{ph}}{2} - \epsilon_0$

Supplementary Table 3 | Phonon-driven spin-Floquet magneto-valleytronics in TMDC bilayer. **a**, The phonon eigenvectors, corresponding to the E'' in the case of monolayer, have two branches in bilayer: E_u and E_g . **b**, The model Hamiltonian for the dichroic behavior of the spin-Floquet states of valleys with respect to a circularly polarized E_u phonon. Note that the spin-Floquet state can be described by the same form as eq. 4 in the main text, once the Δ values are defined, as given in this table.

Supplementary Figure 5 | The time-averaged total spin values of the monolayer and bilayer WTe₂ with an E'' and an E_u phonon mode, respectively. a-b, The cumulative time average of the total spin in monolayer WTe₂ with (a) linear and (b) circular polarized E'' phonon. **c-d**, The same total spin of the bilayer WTe₂ with a circular polarized E_u phonon energy with (c) zero-point phonon ($\frac{1}{2}\hbar\omega_{\text{ph}}$) and (b) single phonon ($\frac{3}{2}\hbar\omega_{\text{ph}}$).

Supplementary Discussion 8. Spin-Floquet magneto-valleytronics in bilayers of TMDC.

We extended the model Hamiltonian study of spin-Floquet magneto-valleytronics to the cases of TMDC bilayers. The E'' phonons in the bilayer have two branches, E_u and E_g , which are depicted in Supplementary Table 3a. Details of the bilayer phonon structures can be

found in a recent literature.^{S1} We focus on the coupling of the CBM spinors to E_u phonon which is known to be IR-active. There are four valleys in the case of the bilayer: K and K' in each of upper and lower layers. The spin on the K point of both the upper and lower layer experience the same in-plane magnetic field, in response to a E_u phonon, while the out of plane magnetic fields of them are directed oppositely. As explained in the case of monolayer, the time-reversal partners (the spin on K' point) in each layer are subjected to exactly opposite direction of in-plane and out-of-plane magnetic field. As a result, the spin-up states (that of K in the upper layer and that of K' in lower layer) evolves with the left-handed Hamiltonian, whereas the spin-down states (that of K' in the upper layer and K in the lower layer) experiences the right-handed Hamiltonian, as given in Supplementary Table.3. The dichroic behavior of the circularly polarized E_u phonon can be parametrized by the Δ parameters. For the same phonon with the opposite circular polarity, the parameters for the four valleys need to be interchanged: Δ_R to Δ_L and vice versa. The same study revealed that the E_g phonon drives the spin in the upper layer exactly opposite to those in the lower layer, keeping the time-reversal symmetry at all time. However, this uninteresting E_g mode is not IR-active, and thus a real experiment can excite only E_u avoiding E_g .

The time-averaged total spin of the bilayer WTe₂ is summarized in Supplementary Fig. 5. As a result of the interplay between the opposite spin direction between the layers and the opposite effective magnetic field between the time-reversal partners, the bilayer produces almost twice increased S_z value while the noisy in-plane component is cancelled in the time-averaged profile, as shown in Supplementary Fig. 5c and 5d.

Here, we would also like to summarize the features of the electronic structure of the bilayer. A distinct character of the bilayer (or a thicker layer) is the emergence of the indirect nature in the band gap. The VBM locates obviously on the Γ -point as a result of interlayer

hybridization. On the other hand, the structure of CBM of the bilayer is not so obvious. Whether the energy minimum is on $K(K')$ or Σ depends on stacking configuration and also computational method. For example, a recent GGA+GW calculation reported that all 2H-polytype TMDC bilayers have its CBM minimum on Σ .^{S2} In the same literature, on the other hand, it was shown that heterogeneous layers (such as $\text{MoS}_2/\text{MoSe}_2$, $\text{MoSe}_2/\text{MoTe}_2$, WSe_2/WS_2 , $\text{WSe}_2/\text{WTe}_2$) have their CBM on $K(K')$. Another previous study using HSE06 hybrid functional reported that heterogeneous trilayers have their CBMs on $K(K')$.^{S3} Thus, there are numerous combinations of the bilayer on which the phonon driven spin-Floquet magnetism can be realized. Even when the CBM locates on Σ , our model for the spin-Floquet valley magnetism is still valid because the spins on the CBM of Σ is as rigid as the VBM of K . To prove this, we calculated the variation of the spin direction of the CBM of K , CBM of Σ , and VBM of K , as summarized in the Supplementary Fig. 6. It shows that the spins at CBM of Σ and VBM of K remain near the equilibrium direction irrespective of the lattice displacement.

[Question 4]

4) I think there is an error in extended data (supplementary) table 1 (second row, third column): the atomic displacement should be in the y-direction.

[Our answer]

We are grateful for the reviewer's careful reading. We corrected it accordingly.

[Revised part of the Supplementary Table 1]

Schematic displacement			
Atomic displacement	$\vec{d}_s = (0.1, 0.0, 0.0) \text{ \AA}$	$\vec{d}_s = (0.0, 0.1, 0.0) \text{ \AA}$	$\vec{d}_s = \left(\frac{0.1}{\sqrt{2}}, \frac{0.1}{\sqrt{2}}, 0.0 \right) \text{ \AA}$
Effective magnetic field	$B_{ph}(1.0, 0.0, 0.0)$	$B_{ph}(0.0, 1.0, 0.0)$	$B_{ph}\left(\frac{1}{\sqrt{2}}, \frac{1}{\sqrt{2}}, 0.0\right)$
CBM(K)	$\langle \vec{S} \rangle = (\pm 0.5, 0.0, 0.0)$	$\langle \vec{S} \rangle = (0.0, \pm 0.5, 0.0)$	$\langle \vec{S} \rangle = \left(\pm \frac{0.5}{\sqrt{2}}, \pm \frac{0.5}{\sqrt{2}}, 0.0 \right)$
VBM(K)	$\langle \vec{S} \rangle = (0.0, 0.0, -0.5)$	$\langle \vec{S} \rangle = (0.0, 0.0, -0.5)$	$\langle \vec{S} \rangle = (0.0, 0.0, -0.5)$

[In response to editorial request]

1. The title is shortened as requested.
2. Author's Affiliations are updated.
3. Subheadings are added in results section of the main text.
4. Styles of equations are modified in accordance with the editorial instruction.
5. We added reference 33, 36, 37 and 38 for the revised main text.
6. End note for Data availability is added, indicating the repository in a public domain.

- 7. All the Extended Data are presented in the file of Supplementary information.**
- 8. We changed “Supplementary section” to “Supplementary Discussion” in accordance with the editorial instruction.**

REVIEWERS' COMMENTS:

Reviewer #1 (Remarks to the Author):

I have carefully read the replies of the authors to my previous requests and I am very impressed by the amount of work that they have added. Therefore, I strongly recommend publication of the manuscript in Nature Communications in its current form. I believe that this paper will encourage a wide community of experimentalists in performing this challenging experiment.

Reviewer #2 (Remarks to the Author):

The authors have answered my questions and have made several changes to the manuscript. Therefore, I recommend its publication.

Reviewers' comments:

Reviewer #1 (Remarks to the Author):

I have carefully read the replies of the authors to my previous requests and I am very impressed by the amount of work that they have added. Therefore, I strongly recommend publication of the manuscript in Nature Communications in its current form. I believe that this paper will encourage a wide community of experimentalists in performing this challenging experiment.

[Our answer]

We are greatly pleased by the Reviewer's evaluation and strong recommendation. We appreciate very much for the Reviewer's kind reading and many good suggestions.

Reviewer #2 (Remarks to the Author):

The authors have answered my questions and have made several changes to the manuscript. Therefore, I recommend its publication.

[Our answer]

We are greatly pleased by the Reviewer's evaluation and recommendation. We appreciate very much for the Reviewer's kind reading in the previous versions and many good suggestions there.